# The European Institute of Oncology Thyroid Imaging Reporting and Data System for Classification of Thyroid Nodules: A Prospective Study

**DOI:** 10.3390/jcm11113238

**Published:** 2022-06-06

**Authors:** Elvio De Fiori, Carolina Lanza, Serena Carriero, Francesca Tettamanzi, Samuele Frassoni, Vincenzo Bagnardi, Giovanni Mauri

**Affiliations:** 1Unità di Radiologia Clinico Diagnostica, European Institute of Oncology IRCCS, 20141 Milan, Italy; elvio.de-fiori@ieo.it; 2Postgraduate School of Radiodiagnostics, University of Milan, 20122 Milan, Italy; carolina.lanza@unimi.it; 3Department of Biomedical Sciences, Humanitas University, 20072 Pieve Emanuele, Italy; f.tettamanzi1@campus.unimib.it; 4Department of Statistics and Quantitative Methods, University of Milan-Bicocca, 20126 Milan, Italy; samuele.frassoni@unimib.it (S.F.); vincenzo.bagnardi@ieo.it (V.B.); 5Department of Oncology and Hematology-Oncology, University of Milan, 20122 Milan, Italy; vannimauri@gmail.com; 6Division of Interventional Radiology, European Institute of Oncology IRCCS, 20141 Milan, Italy

**Keywords:** thyroid, thyroid nodule, TIRADS, US, diagnosis, computer-aided diagnosis

## Abstract

Background: To evaluate the performance, quality and effectiveness of “IEO-TIRADS” in assigning a TI-RADS score to thyroid nodules (TN) when compared with “EU-TIRADS” and the US risk score calculated with the S-Detect software (“S-Detect”). The primary objective is the evaluation of diagnostic accuracy (DA) by “IEO-TIRADS”, “S-Detect” and “EU-TIRADS”, and the secondary objective is to evaluate the diagnostic performances of the scores, using the histological report as the gold standard. Methods: A radiologist collected all three scores of the TNs detected and determined the risk of malignancy. The results of all the scores were compared with the histological specimens. The sensitivity (SE), specificity (SP), and diagnostic accuracy (DA), with their 95% confidence interval (95% CI), were calculated for each method. Results: 140 TNs were observed in 93 patients and classified according to all three scores. “IEO-TIRADS” has an SE of 73.6%, an SP of 59.2% and a DA of 68.6%. “EU-TIRADS” has an SE of 90.1%, an SP of 32.7% and a DA of 70.0%. “S-Detect” has an SE of 67.0%, an SP of 69.4% and a DA of 67.9%. Conclusion: “IEO-TIRADS” has a similar diagnostic performance to “S-Detect” and “EU-TIRADS”. Providing a comparable DA with other reporting systems, IEO-TIRADS holds the potential of being applied in clinical practice.

## 1. Introduction

Thyroid nodules (TN) represent a common finding in clinical practice, with an incidence of 19–68% in the general population [1]. The vast majority of TNs are benign (colloid nodule, follicular adenoma, cyst, and thyroiditis), and 5 to 15% are malignant (papillary, follicular, medullary, or anaplastic carcinoma) [2,3]. No single US characteristic, or a blend of US features, is sensitive or specific enough to identify all malignant nodules [4]. Indeed, the differentiation between benign and malignant nodules, in clinical practice, represents the most significant challenge.

Only 5–7% of TNs in the adult population are detected by physical examination; the majority of them are asymptomatic, and are detected incidentally with a prevalence of 65% by conventional ultrasound (US), 15% by computed tomography (CT) or magnetic resonance imaging (MRI), and 1.2% by fluorodeoxyglucose positron emission tomography (PET) [5].

Conventional US is recommended as the first-line evaluation of TNs because it does not require ionizing radiation, is real time, is low cost, can provide a view on different axes, and is largely accessible [4].

Therefore, in recent years, many standardized systems for reporting thyroid US have been developed in an effort to identify nodule characteristics associated with risk of malignancy [6,7,8]. The most common classification system is the Thyroid Imaging Reporting and Data System (TI-RADS), proposed by Horvath et al. in 2009 [9]. TI-RADS is a point scale based on the number and combination of malignancy predictors on US that categorize the TN as low-to-highly suspicious. Several US features have been associated with a higher risk of malignancy, based on nodule composition, echogenicity, shape, margin, and the presence of echogenic foci (comet-tail artifacts, macrocalcification, peripheral calcification, and punctate echogenic foci) [10].

Over the years, TI-RADS classification has been modified and thyroid and imaging society guidelines recommended different versions: EU-TIRADS (provided by the European Thyroid Association) [11], K-TIRADS (by the Korean Society of Thyroid Radiology) and ACR-TIRADS (by the American College of Radiology) [12,13]. All these TI-RADS versions demonstrated excellent diagnostic value in predicting thyroid malignancy [14,15].

A new generation of US machines, computer-aided diagnosis (CAD) and other stratification systems can improve US image evaluation. New US software has been developed to automatically assess the US features and evaluate the TN in terms of the degree of risk of malignancy after the identification of a TN by the operator [16]. One of these pieces of software, “S-Detect”, is used to evaluate the risk of malignancy of TNs, and automatically assesses the following US features: composition, margins, echogenicity, orientation, spongiform appearance, shape, and calcifications.

Nevertheless, the diagnosis of malignancy of TNs cannot exist regardless of the cytological examination. Fine-needle aspiration biopsy (FNAB) has an essential role in evaluating malignant and benignant TNs. FNAB in expert centers provides valid results in 65–75% of examined nodules [17]. Of them, almost 60–70% are benign, 5% are positive for papillary carcinoma, 5–15% are inconclusive, and the remaining 15–25% of aspirates are indeterminate or suspicious [18]. However, cytological sampling with FNAB has intrinsic limitations, so the final diagnostic is achieved with analysis of the surgical specimen in operated patients.

In this study, we propose a new and simplified model of a standardized US report, developed at our institution, the European Institute of Oncology, named “IEO-TIRADS”, which includes a numerical score and a graphical representation of the thyroid gland and cervical lymph node (LN) levels.

This model is based on the study of the main articles on TNs and the work experience at our institute [19,20].

The aim of this prospective study is to evaluate the performance, quality and effectiveness of “IEO-TIRADS” in assigning a TI-RADS score to a TN when compared with both the EU-TIRADS, named “EU-TIRADS”, and the standard US risk score calculated with S-Detect software, named “S-Detect”.

The primary objective will be the evaluation of the diagnostic accuracy (DA) by “IEO-TIRADS”, “S-Detect” and “EU-TIRADS”, using the cytological report to confirm the diagnosis. The hypothesis is that the DA of the “IEO-TIRADS” score is similar to that of both “S-Detect” and “EU-TIRADS”. The secondary objective is to evaluate the diagnostic performances in terms of sensitivity (SE) and specificity (SP) of “IEO-TIRADS”, “S-Detect” and “EU-TIRADS”, as compared with the cytological reports.

## 2. Materials and Methods

### 2.1. Study Design

This study was conducted with approval from the local ethics committee, and written informed consent was obtained from all 93 patients enrolled prior to inclusion.

First, a radiologist detects the lesion with 2D-mode US, determines the risk of malignancy, and collects the scores of “S-Detect”, “IEO-TIRADS” and “EU-TIRADS”. “S-Detect” was collected using the “S-Detect” standard scoring system already set in Samsung Ultrasound SRS80 Prestige. Next, the same radiologist generates a graphical representation, which maps the TN and cervical LN identified during the US examination to evaluate “IEO-TIRADS”.

Finally, the results of all radiologist’s scores will be compared with the histological result of the surgical specimen.

### 2.2. Patients

From 2018 to 2020, 93 patients (61 women and 32 men) already scheduled for thyroid surgery with 140 TNs were prospectively enrolled in the study.

The inclusion criteria were as follows: (1) adults (age ≥ 18 years) of both sexes between 18 and 80 years old with TN candidates for thyroid surgery regardless of histologic results; (2) the patient and the investigator have to sign written informed consent prior to inclusion.

The exclusion criteria were as follows: (1) patients who refuse to undergo surgery and/or to participate in the study; (2) patients with nodules incorrectly parametrizable; (3) non-operable patients.

### 2.3. Data Acquisition

All patients gave their informed consent before inclusion in the study.

Images were obtained using a 3–12 Hz linear-array transducer on a Samsung RS80A. The scanning protocol includes scanning of the thyroid gland and cervical LN in both transverse and longitudinal planes by brightness mode (B-mode), color-coded Doppler imaging (CCDI) and power Doppler imaging (PDI). Images and TI-RADS evaluations were collected and compared. All US patterns and descriptions of features were available and analyzed by a radiologist with 15 years of experience in thyroid imaging.

To evaluate “IEO-TIRADS”, the following data for any nodes were collected: (a) site of the lesion (right or left); (b) WxH (width x height); (c) lobe of the lesion; (d) level of the lesion; (e) morphology; (f) echostructure; (g) vascularization; (h) contact with capsule (yes (+1) or no); and (f) the presence of pathological LN (yes (+1) or no) (Figure 1, Figure 2 and Figure 3).

The scoring systems for morphology, echostructure and vascularization are listed in Table 1.

This scheme allows us to scrupulously consider the TN’s US aspects and to choose the most suitable TI-RADS score. In “IEO-TIRADS”, clinically significant disease is categorized as follows: 1—very unlikely (3 points), 2—unlikely (4–6 points), 3—doubt (7–9 points), 4—suspected (10–12 points), or 5—malignant (13–15 points) (Table 2).

To obtain “EU-TIRADS”, the criteria of EU-TIRADS were followed, evaluating the following fundamental US features: shape, margins, echogenicity, composition, and hyperechoic foci [11].

The “S-Detected” score was calculated automatically by using the “S-Detect” standard scoring system, with the model coming from the three major imaging and data reporting systems (K-TIRADS, Russ, and ATA guidelines) already set in Samsung Ultrasound SRS80 Prestige [21,22].

### 2.4. Statistical Analysis

Continuous data were reported as median and ranges. Categorical data were reported as counts and percentages. The accuracy of defining the malignant or benign nature of the nodules of “IEO-TIRADS”, “EU-TIRADS” and “S-Detect” was evaluated using the histological report as the gold standard.

SE, SP, and DA, with their 95% confidence interval (95% CI), were calculated for each of the three methods.

The primary endpoint of the study was DA, defined as the number of correct assessments of the nodules divided by the number of all assessments. The 95% CIs for the estimated DA, SE, and SP differences between “IEO-TIRADS” and “S-Detect” and among “IEO-TIRADS” and “EU-TIRADS” were calculated using the Wald method with Bonett–Price Laplace adjustment [23]. The McNemar test was applied to evaluate the significance of the differences. All analyses were performed with the statistical software SAS 9.4 (SAS Institute, Cary, NC, USA). A *p*-value less than 0.05 was considered statistically significant.

## 3. Results

One-hundred and forty TNs were observed in 93 patients and classified according to “IEO-TIRADS”, “EU-TIRADS” and “S-Detect”. The median age of patients was 48 years (range 22–81).

Among these patients, 58 (62%) presented a single nodule, 24 (26%) two nodules, 10 (11%) three nodules and, finally, 1 patient four nodules (1%).

The histological report detected 49 benign TNs (35%) and 91 malignant TNs (65%). Histological data are reported in Table 3. The lesion classification, SE, SP and DA of the three methods are reported in Table 4 (see Appendix A for sensitivity, specificity and diagnostic accuracy according to different cutoff of IEO-TIRADS score).

According to “IEO-TIRADS”, 67 of 91 nodules were correctly classified as malignant, resulting in an SE of 73.6% [95% CI: 64.6–82.7%]; 29 of 49 nodules were correctly classified as benign, leading to an SP of 59.2% [95% CI: 45.4–73.0%]. The DA was 68.6% [95% CI: 60.9–76.3%].

According to “EU-TIRADS”, 82 of 91 nodules were correctly classified as malignant, reaching an SE of 90.1% [95% CI: 84.0–96.2%]; 16 of 49 nodules were correctly classified as benign, resulting in an SP of 32.7% [95% CI: 19.5–45.8%]. The DA was 70.0% [95% CI: 62.4–77.6%].

According to “S-Detect”, 61 of 91 nodules were correctly classified as malignant, with an SE of 67.0% [95% CI: 57.4–76.7%]; 34 of 49 nodules were correctly classified as benign, resulting in an SP of 69.4% [95% CI: 56.5–82.3%]. The DA was 67.9% [95% CI: 60.1–75.6%].

The difference between the DA among the “IEO-TIRADS” and “S-Detect” methods, and among “IEO-TIRADS” and “EU-TIRADS” was, respectively, 0.7% [95% CI: −8.3–9.8%] and −1.4% [95% CI: -9.7–6.9%]. The estimated differences [95% CI] and *p*-values of “IEO-TIRADS” vs. “S-Detect” and “IEO-TIRADS” vs. “EU-TIRADS” are reported in Table 5 (see Supplementary Material for ROC curve of the model with IEO-TIRADS score as independent variable and histopathological result (Benign vs. Malignant) as dependent variable).

## 4. Discussion

Since 2009, when Horvath and Park initially proposed TI-RADS, the scoring system has been modified over the years, and it is ongoing evaluation and being amended to improve the accuracy of the diagnosis of malignant TNs with US [24]. Nowadays, different TIRADS systems have been proposed, but are still not widely applied to the perceived complexity of the reporting system in clinical practice [25].

US has a pivotal role in the diagnosis of TNs. However, US alone has several limitations regarding the overlapping boundaries, morphology, vascularization and echostructure. Moreover, the accuracy of diagnosis is affected by subjective factors related to the radiologist, because US is an operator-dependent exam.

The diagnosis of TNs directly affects the therapeutic decisions and patient prognosis; therefore, it has important clinical significance.

Thyroid nodule management has been modified significantly in recent years, with the inclusion of observation protocols for low-risk thyroid cancer and minimally invasive treatments such as thermal ablation for both benign and malignant thyroid nodules [26,27,28].

The improvement of a simpler reporting system would help to achieve larger diffusion of standardized reporting, with a consequential relevant impact on the complex clinical management of patients with thyroid nodules. In this scenario, the so-called IEO-TIRADS has been developed in accordance with radiologists and surgeons at our institution, to provide a simple and reliable reporting system.

This study’s aim is to compare the diagnostic performance of “IEO-TIRADS”, “S-Detect” and “EU-TIRADS” for predicting the risk of TN malignancy. To the best of our knowledge, this study is the only study that compares the diagnostic performance of “S-Detect” software, “EU-TIRADS” and “IEO-TIRADS” classification in predicting TN malignancy.

S-Detect software is an advanced innovative device, with the aim of improving the non-invasive classification of TNs [16,25,29,30]. S-Detect is an available software and has been used in previous studies [25].

In our study, EU-TIRADS showed higher SE but lower SP, while S-Detect had higher SP but lower SE. IEO-TIRADS, on the contrary, achieved good results in both SE and SP. The data of our study are quite consistent with other recent studies. Dobruch-Sobczak et al., have reported 98.7% SE and 39.8% SP [26]. Different data are obtained by Schenke et al., with an SE of 49.3% and an SP of 97.4% [27,28].

The primary aim of this study was to determine whether the accuracy of defining the malignant or benign nature of the nodules of “IEO-TIRADS” was similar to that of both “S-Detect” and “EU-TIRADS”, using the histological report as the gold standard.

The DAs of “IEO-TIRADS”, “S-Detect” and “EU-TIRADS” were, respectively, 68.6% [95% CI: 60.9–76.3%], 67.9% [95% CI: 60.1–75.6%] and 70.0% [95% CI: 62.4–77.6%].

The difference between the DA among “IEO-TIRADS” and “S-Detect” methods, and among “IEO-TIRADS” and “EU-TIRADS” was, respectively, 0.7% [95% CI: −8.3–9.8%] and −1.4% [95% CI: −9.7–6.9%], suggesting that the diagnostic accuracy of the “IEO-TIRADS” method was similar to that of both “S-Detect” and “EU-TIRADS”.

Our study has some limitations. The first is the oncological population of our study, because the performance of different classification systems depends on the population being examined. Second is the small cohort of TNs. The third is that this study was conducted in a single center, and all US examinations were performed by a radiologist with extensive experience, so intra and interobserver agreement were not evaluated in this study.

We can conclude that the “IEO-TIRADS” classification system for the evaluation of TNs has a similar diagnostic performance to both “S-Detect” and “EU-TIRADS”.

We believe that this new model makes the report easier to read and understand by surgeons and patients. The graphical representation maps of TNs and cervical LNs identified during the US examination facilitate the TN and LN location according to anatomical planes. This increases the diagnostic confidence in the evaluation of the benign or malignant nature of a lesion in a rapid, easy and effective way.

This new system is simple to be applied and is also appreciated by surgeons and endocrinologists. Providing comparable DA with other reporting systems, IEO-TIRADS holds the potential of being largely applied in clinical practice.

## Figures and Tables

**Figure 1 jcm-11-03238-f001:**
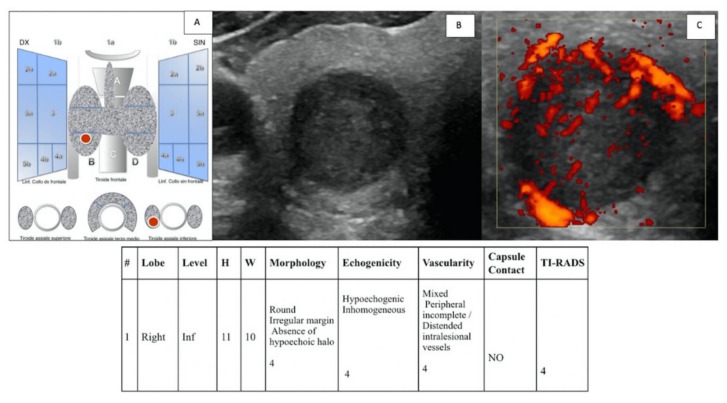
(**A**) “IEO-TIRADS” graphical representation of solid TN in the right thyroid lobe, lower lobe. US images with B-mode (**B**) and CCDI (**C**) demonstrate the presence of TN of 11 × 10 mm (HxW) with round irregular margin, absence of hypoechoic halo, inhomogeneous hypoechogenicity, mixed peripheral vascularization and incomplete/distended intralesional vessels, and no contact with capsule (all scoring data are collected in the table). A TI-RADS 4 score was assigned, and the histological examination revealed a papillary carcinoma (pT1b pN0).

**Figure 2 jcm-11-03238-f002:**
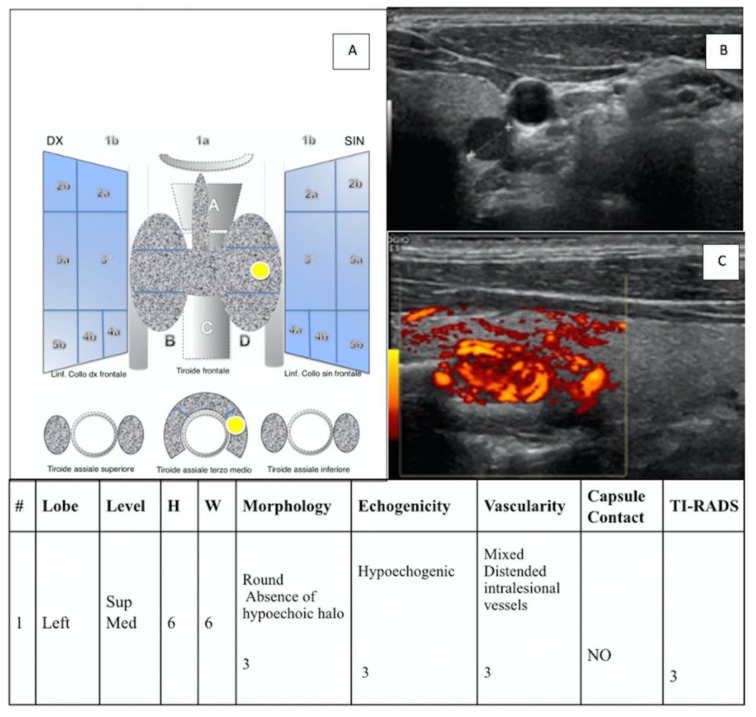
(**A**) “IEO-TIRADS” graphical representation of solid TN in the left thyroid lobe, middle lobe. US images with B-mode (**B**) and CCDI (**C**) demonstrate the presence of TN of 6 × 6 mm (HxW) with round irregular margin, absence of hypoechoic halo, hypoechogenicity, mixed vascularization and distended intralesional vessels, and no contact with capsule (all scoring data are collected in the table). A TI-RADS 3 score was assigned, and the histological examination revealed a follicular adenoma fetal type.

**Figure 3 jcm-11-03238-f003:**
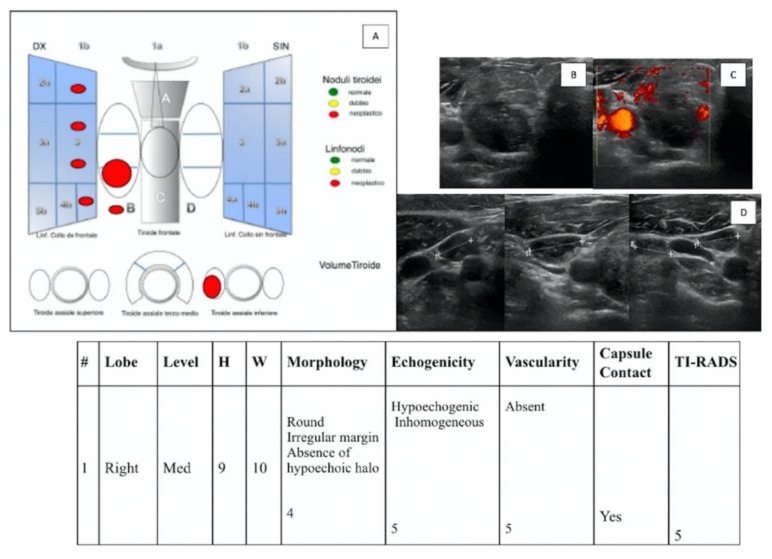
(**A**) “IEO-TIRADS” graphical representation of solid TN in the thyroid middle-inferior right lobe. US images with B-mode (**B**) and CCDI (**C**) demonstrate the presence of TN of 9 × 10 mm (HxW), round, with irregular margin, absence of hypoechoic halo, inhomogeneous hypoechogenicity, absence of vascularization, and contact with capsule (all scoring data are collected in the table). The presence of lymph nodes with “malignant” features at levels IIa, III, and Iva on the right side was also demonstrated (**D**). A TI-RADS 5 score was assigned, and the histological examination revealed a papillary carcinoma (pT1b, pN1b).

**Table 1 jcm-11-03238-t001:** The “IEO-TIRADS” scoring systems for morphology, echostructure and vascularization.

Morphology	1 = wider than high 2 = round shape 3 = higher than wide +1 irregular-shade margin +1 absence of hypoechoic halo
Echostructure	1 = hyperechogenic 2 = iso—hypoechogenic 3 = hypoechogenic +1 inhomogeneous +1 hyperechoic spots/microcalcification
Vascularity	1 = peripheral 2 = mixed 3 = peripheral with intralesional radiations 5 = absent +1 peripheral and incomplete +1 distended intralesional vessels

**Table 2 jcm-11-03238-t002:** The “IEO-TIRADS” scoring system cut off to classify the nodule as benign or malignant.

IEO-TIRADS Classification	IEO-TIRADS Score	Frequency	Percent
Benign	1 (Very unlikely)	9	6.43
Benign	2 (Unlikely)	44	31.43
** *Malignant* **	** *3 (Doubt)* **	** *23* **	** *16.43* **
Malignant	4 (Suspected)	26	18.57
Malignant	5 (Malignant)	38	27.14

**Table 3 jcm-11-03238-t003:** Data of final histological diagnosis.

Histological Types	Number of Patients
**FOLLICULAR CARCINOMA**	12
Papillary carcinoma	71
**MEDULLARY CARCINOMA**	2
Nodular hyperplasia	25
**FOLLICULAR ADENOMA**	30

**Table 4 jcm-11-03238-t004:** Lesion classification, sensitivity, specificity and diagnostic accuracy of the three methods.

		Histopathological Lesion Result			Diagnostic Performance		
Method	Lesion	Benign	Malignant	Total	Sensitivity [95% CI]	Specificity [95% CI]	Accuracy [95% CI]
IEO-TIRADS	Benign	29	24	53	73.6% [64.6–82.7%]	59.2% [45.4–73.0%]	68.6% [60.9–76.3%]
	Malignant	20	67	87			
S-Detect	Benign	34	30	64	67.0% [57.4–76.7%]	69.4% [56.5–82.3%]	67.9% [60.1–75.6%]
	Malignant	15	61	76			
EU-TIRADS	Benign	16	9	25	90.1% [84.0–96.2%]	32.7% [19.5–45.8%]	70.0% [62.4–77.6%]
	Malignant	33	82	115			
Total		49	91	140			

**Table 5 jcm-11-03238-t005:** Estimated differences and *p*-values of “IEO-TIRADS” vs. “S-Detect” and “IEO-TIRADS” vs. “EU-TIRADS”.

Methods	Estimated Difference [95% CI] and *p*-Value		
	Sensitivity	Specificity	Diagnostic Accuracy
**“IEO-TIRADS” vs.** **“S-Detect”**	6.6% [95% CI: −5.4–18.3%], *p* = 0.27	−10.2% [95% CI: −23.4–3.8%], *p* = 0.13	0.7% [95% CI: −8.3–9.8%], *p* = 0.88
**“IEO-TIRADS” vs. “EU-TIRADS”**	−16.5% [95% CI: −25.2–−7.0%], *p* < 0.001	26.5% [95% CI: 11.3–39.7%], *p* < 0.001	−1.4% [95% CI: −9.7–6.9%], *p* = 0.73

## Data Availability

The data presented in this study are available on request from the corresponding author.

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
