# Peer review of "The European Institute of Oncology Thyroid Imaging Reporting and Data System for Classification of Thyroid Nodules: A Prospective Study"

_jcm, 2022, doi:10.3390/jcm11113238_

Round 1
Reviewer 1 Report
This is a study about the evaluation of diagnostic performance of IEO-TIRADS score to assess the risk of malignancy of Thyroid nodule (TN). This score is compared to the results obtaines with EU-TIRADS and a S-Detect Software (“S-Detect”). This is an interesting topic. The graphical representation associated with the IEO-TIRADS” can provide a better visual information about the TNs, which can be helpful for discussion with the surgeon or for US follow up.
However, there are some problems with the study that need to be clarified
Majors issues :
Regarding the malignancy criteria used:
- Hypoechogenicity is not graded whereas marked hypoechogenicity is a strong criterion of malignancy (EU-TIRADS 5) in contrast to moderate or mild hypoechogenicity which is a minor criterion (EU-TIRADS 4 in the absence of strong criteria of malignancy)
- It is not noted how the classification system handles hyperechogenic spots? either for the IEO-TIRADS score (I think it is up to the operator to make the difference as for EU-TIRADS but it is not at all clear in table 1) or the S-Detect. Do the systems reach to classify microcalcifications versus inspissated colloid (comet-tail artifact)?
Regarding the gold standard and statistical analysis
- The assignment of a value (between 1 and 5) according to the score is not clear and should be explained.
- There are errors in the text, the gold standard here is histological (surgery) and not cytological.
- Which cut-off score did you use to classify the nodule as benign or malignant (EU TIRADS≥4 as malignant I presume, but for the others?).
- What is the difference between the scores in terms of sensitivity or specificity? (in other words, why the results are different, which criteria are involved ?) These data should be analyzed and presented to the reader.
- There is no information about the comparison between cytological/histological data and the different classification…There is no data about the final histological diagnosis..
Regarding the internal and external validity of these scores, there is no assessment of intra- and inter-operator variability for each criterion and each classification. From memory, vascularization was not included in the TIRADS classification because of the problem of inter-observer variability. What about here for the IEO-TIRADS ? Same question regarding the classification of hyperechogenic spots (see above) or invasion of the capsule in ultrasound, is it a reproducible parameter?
Minors comments :
Introduction
Page 3 line 57. There are other discovery circumstances (TN detected incidentally by conventional imaging (i.e US, TDM, MRI) or fonctionnal imaging, such PET /CT). Please the authors reformulate the sentence.
Page 3 lines 62-64 should be put before 59-61
Page 3 line 78 « Thyroid US is the gold standard for TN’s detection and cancer risk stratification of TN » This sentence is repetitive and should be deleted
Page 3 line 82 « One of these softwares
Page 3 line 90 « highest DA » should be replaced by « final diagnostic »
Page 4 : « However, even cytological sampling with FNAB can provide a wrong diagnosis, and the highest DA is achieved with analysis of the surgical specimen in operated patients »
It is not necessarily appropriate to speak of a misdiagnosis. Cytological analysis simply has limitations due to the fact that it is a cytological analysis. Please rephrase.
Page 4 line 96-100 : This part should be report in the discussion
Page 4 lines 104-105 : cytological is not the gold standard in the study (the correct formulation is histological report). This mistake is present in other places in the text
Material and methods
Page 5 line 125 : « the study »
Page 5 « (1) adults (age≥18 years) of both sex between 18-80 years old
127 with TN candidates to thyroid surgery regardless of cytologic results. » Can the authors provide the surgical indication ? (suspicion of malignancy ?, goiter ? hyperthyroidism ?)
Results
Page 7 line 177 « Hystological …. »
Discussion
Page 8 lines 220 221
Reference 24 showed that the CAD evaluation of thyroid nodules stratification risk has a potential role in a didactic field and does not play a real and effective role in the clinical field, where not only images but also specialistic medical practice is fundamental to achieve a diagnosis based on family history, genetics, lab tests, and so on. The CAD system may be useful for less experienced operators as its specificity was significantly higher.
So we can't talk about clinical validation in my opinion...
Reference 25 is for breast cancer…
Author Response
Reviewer 1
Comments and Suggestions for Authors
This is a study about the evaluation of diagnostic performance of IEO-TIRADS score to assess the risk of malignancy of Thyroid nodule (TN). This score is compared to the results obtains with EU-TIRADS and a S-Detect Software (“S-Detect”). This is an interesting topic. The graphical representation associated with the IEO-TIRADS” can provide a better visual information about the TNs, which can be helpful for discussion with the surgeon or for US follow up.
However, there are some problems with the study that need to be clarified
Majors issues:
Regarding the malignancy criteria used:
- Hypoechogenicity is not graded whereas marked hypoechogenicity is a strong criterion of malignancy (EU-TIRADS 5) in contrast to moderate or mild hypoechogenicity which is a minor criterion (EU-TIRADS 4 in the absence of strong criteria of malignancy)
Yes, to make the score simple we only used hypoecogenicity without distinction between marked and moderate hypoecogenicity.
- It is not noted how the classification system handles hyperechogenic spots? either for the IEO-TIRADS score (I think it is up to the operator to make the difference as for EU-TIRADS but it is not at all clear in table 1) or the S-Detect. Do the systems reach to classify microcalcifications versus inspissated colloid (comet-tail artifacts)
The presence of microcalcifications was noted by the operator, distinguishing from inspissated colloid. The evaluation of the ability of SDetect to distinguish those two features was beyond the aim of the present paper.
Regarding the gold standard and statistical analysis
- The assignment of a value (between 1 and 5) according to the score is not clear and should be explained.
Values has been better explained in the text.
- There are errors in the text, the gold standard here is histological (surgery) and not cytological.
Errors have been corrected.
- Which cut-off score did you use to classify the nodule as benign or malignant (EU TIRADS≥4 as malignant I presume, but for the others?).
We only assigned a score to each nodule according to his US features, and then compared it with final histological results, which was used to classify nodules as benign or malignant.
- What is the difference between the scores in terms of sensitivity or specificity? (in other words, why the results are different, which criteria are involved ?) These data should be analyzed and presented to the reader.
The reported difference between DAs was the simple algebraic difference “A-B”. As reported in the methods section, the 95% CIs for the estimated DA difference between “IEO-TIRADS” and “S-Detect” and among “IEO-TIRADS” and “EU-TIRADS” were calculated using the Wald method with Bonett–Price Laplace adjustment [Bonett DG, Price RM. Adjusted Wald Confidence Interval for a Difference of Binomial Proportions Based on Paired Data. J Educ Behav Stat. 2012;37(4):479-488.]
To test the statistical significance of the differences between two methods, the McNemar test was now applied, and p-values reported
- There is no information about the comparison between cytological/histological data and the different classification…There is no data about the final histological diagnosis..
Comparison between cytological and histological diagnosys was beyond the aim of this work. We only included a selected group of patients who underwent surgery in order to have a more robust final diagnosis. So this would be not the ideal sample to make a comparison between cytological and histological analysis. We had add data about the final histological diagnosis.
Regarding the internal and external validity of these scores, there is no assessment of intra- and inter-operator variability for each criterion and each classification. From memory, vascularization was not included in the TIRADS classification because of the problem of inter-observer variability. What about here for the IEO-TIRADS ? Same question regarding the classification of hyperechogenic spots (see above) or invasion of the capsule in ultrasound, is it a reproducible parameter?
All the US images were assessed by the same experienced readers, and intra and inter-observer agreement were not evaluated. This has been added in the limitation section of the study.
Minors comments:
Introduction
Page 3 line 57. There are other discovery circumstances (TN detected incidentally by conventional imaging (i.e US, TDM, MRI) or fonctionnal imaging, such PET /CT). Please the authors reformulate the sentence.
Thanks for the comment, we have modified the sentence. “Only 5-7% of TN in the adult population are detected by physical examination, the most part of them are asymptomatic detected incidentally with a prevalence of 65% by conventional ultrasound (US), 15% by computed tomography (CT) or magnetic resonance imaging (MRI) and 1,2% detected by fluorodeoxyglucose positron emission tomography (PET).”
Page 3 lines 62-64 should be put before 59-61
Thanks for the comment, we have modified the sentence. “No single US characteristic or a blend of US features is sensitive or specific to identify all malignant nodules [4]. Indeed, the differentiation between benign and malignant nodules, in clinical practice, represents the most significant challenge. Only 5-7% of TN in the adult population are detected by physical examination, the most part of them are asymptomatic detected incidentally with a prevalence of 65% by conventional ultrasound (US), 15% by computed tomography (CT) or magnetic resonance imaging (MRI) and 1,2% detected by fluorodeoxyglucose positron emission tomography (PET).”
Page 3 line 78 « Thyroid US is the gold standard for TN’s detection and cancer risk stratification of TN » This sentence is repetitive and should be deleted
Thanks for the comment, we have modified the sentence.
Page 3 line 82 « One of these softwares.
Thanks for the comment, we have modified the sentence. “One of these software, the “S-Detect” software…”
Page 3 line 90 « highest DA » should be replaced by « final diagnostic »
Thanks for the comment, we have modified the sentence. “However, the cytological sampling with FNAB has intrinsic limitations so the final diagnostic is achieved with analysis of the surgical specimen in operated patients.”
Page 4 : « However, even cytological sampling with FNAB can provide a wrong diagnosis, and the highest DA is achieved with analysis of the surgical specimen in operated patients »
It is not necessarily appropriate to speak of a misdiagnosis. Cytological analysis simply has limitations due to the fact that it is a cytological analysis. Please rephrase.
Thanks for the comment, we have rephrased the sentence. “However, the cytological sampling with FNAB has intrinsic limitations so the final diagnostic is achieved with analysis of the surgical specimen in operated patients.”
Page 4 line 96-100 : This part should be report in the discussion
Thanks for the comment, we have report in the discussion the sentence. “…We can conclude that “IEO-TIRADS” classification system for the evaluation of TN has a similar diagnostic performance to both “S-Detect” and “EU-TIRADS”. We believe that this new model makes the report easier to read and understand by surgeons and patients. The graphical representation maps of TN and cervical LN identified during the US examination facilitate the TN and LN location according to anatomical planes. This increases the diagnostic confidence in evaluating of the benign or malignant nature of a lesion in a rapid, easy and effective way….”
Page 4 lines 104-105 : cytological is not the gold standard in the study (the correct formulation is histological report). This mistake is present in other places in the text
Thanks for the comment, in this study we have used the histological report to confirm the diagnosis.
Material and methods
Page 5 line 125 : « the study »
Thanks for the comment, we have modified the sentence.
Page 5 « (1) adults (age≥18 years) of both sex between 18-80 years old
127 with TN candidates to thyroid surgery regardless of cytologic results. » Can the authors provide the surgical indication ? (suspicion of malignancy ?, goiter ? hyperthyroidism ?)
We are sorry, but we found difficulties in clearly identify the indication to surgery for each patient, that derived from multidisciplinary discussion and took into consideration several aspects. So we cannot provide those data
Results
Page 7 line 177 « Hystological …. »
Thanks for the comment, we have modified the sentence.
Discussion
Page 8 lines 220 221
Reference 24 showed that the CAD evaluation of thyroid nodules stratification risk has a potential role in a didactic field and does not play a real and effective role in the clinical field, where not only images but also specialistic medical practice is fundamental to achieve a diagnosis based on family history, genetics, lab tests, and so on. The CAD system may be useful for less experienced operators as its specificity was significantly higher.
So we can't talk about clinical validation in my opinion...
Reference 25 is for breast cancer…
Thanks for the comment, we have modified the sentence and eliminates ref.25. “S-Detect is an available software and has been used in previous studies [24]….”

Reviewer 2 Report
In this study, the authors developed a novel assessment system, IEO-TIRADS, to evaluate the risk of malignancy of thyroid nodule based on ultrasound parameters. Comparing with existed “EU-TIRADS” and ‘’S-Detect” system, the IEO-TIRADS indicated similar diagnostic performance and might be applied in clinical practice. The IEO-TIRADS scoring system collects morphology, echostructure and vascularity data from ultrasound images, which makes it simple for clinical application. However, several points need to be noticed and further discussed.
- In Introduction section the author mentioned’ Thyroid US is the gold standard for TN’s detection and cancer risk stratification of TN”. Please cite literatures to support this statement.
- In Materials and Methods section, the inclusion and exclusion criteria are not rigorous enough. Did the cohort include or exclude patients with previous therapeutic history of thyroid cancer? Also, the clinicopathological data of the cohort should be added and listed as a Table in this study.
- In Materials and Methods section, the author mentioned “using the cytological report as the gold standard”. As the cytological exam (fine needle-aspiration biopsy) provides 65-75% valid results (Line 87-88) and patients in this cohort were scheduled for surgery, why not use the pathological report from surgical specimen as the gold standard?
- In Results section, the author mentioned “The difference between the DA among IEO-TIRADS and S-Detect method, and among IEO-TIRADS and EU-TIRADS was, respective, 0.7% and -1.4%”. Can we calculate the differences between two methods by just A minus B? Did the differences of DA between two groups were statistical significant? This part should be clarified.
- In Discussion section, the author indicated the clinical performance of IEO-TIRADS was similar to other TIRADS system. It is recommended to discuss more advantages of IEO-TIRADS than EU-TIRADS and S-Detect, which strengthens the purpose of this study.
Author Response
Reviewer 2
Comments and Suggestions for Authors
In this study, the authors developed a novel assessment system, IEO-TIRADS, to evaluate the risk of malignancy of thyroid nodule based on ultrasound parameters. Comparing with existed “EU-TIRADS” and ‘’S-Detect” system, the IEO-TIRADS indicated similar diagnostic performance and might be applied in clinical practice. The IEO-TIRADS scoring system collects morphology, echostructure and vascularity data from ultrasound images, which makes it simple for clinical application. However, several points need to be noticed and further discussed.
- In Introduction section the author mentioned’ Thyroid US is the gold standard for TN’s detection and cancer risk stratification of TN”. Please cite literatures to support this statement.
Thanks for the comment, we have deleted the sentence because was repetitive.
- In Materials and Methods section, the inclusion and exclusion criteria are not rigorous enough. Did the cohort include or exclude patients with previous therapeutic history of thyroid cancer? Also, the clinicopathological data of the cohort should be added and listed as a Table in this study.
Yes, no patients with previous history of thyroid cancer were included in the present series. Final data about histological diagnosis have been added to the text.
- In Materials and Methods section, the author mentioned “using the cytological report as the gold standard”. As the cytological exam (fine needle-aspiration biopsy) provides 65-75% valid results (Line 87-88) and patients in this cohort were scheduled for surgery, why not use the pathological report from surgical specimen as the gold standard?
This was a taping mistake. Histological data were used, and this is one of the strengths of the present study.
- In Results section, the author mentioned “The difference between the DA among IEO-TIRADS and S-Detect method, and among IEO-TIRADS and EU-TIRADS was, respective, 0.7% and -1.4%”. Can we calculate the differences between two methods by just A minus B? Did the differences of DA between two groups were statistical significant? This part should be clarified.
The reported difference between DAs was the simple algebraic difference “A-B”. As reported in the methods section, the 95% CIs for the estimated DA difference between “IEO-TIRADS” and “S-Detect” and among “IEO-TIRADS” and “EU-TIRADS” were calculated using the Wald method with Bonett–Price Laplace adjustment [Bonett DG, Price RM. Adjusted Wald Confidence Interval for a Difference of Binomial Proportions Based on Paired Data. J Educ Behav Stat. 2012;37(4):479-488.]
To test the statistical significance of the differences between two methods, the McNemar test was now applied, and p-values reported.
“The primary endpoint of the study was DA, defined as the number of correct assessments of the nodules divided by the number of all assessments. The 95% CIs for the estimated DA, SE, and SP differences between “IEO-TIRADS” and “S-Detect” and among “IEO-TIRADS” and “EU-TIRADS” were calculated using the Wald method with Bonett–Price Laplace adjustment [22]. The McNemar test was applied to evaluate the significance of the differences. All analyses were performed with the statistical software SAS 9.4 (SAS Institute, Cary, NC). A p-value less than 0.05 was considered statistically significant.”
Methods |
Estimated difference [95% CI] and P-value |
||
Sensitivity |
Specificity |
Diagnostic accuracy |
|
“IEO-TIRADS” vs. “S-Detect” |
6.6% [95% CI: -5.4% - 18.3%], P=0.27 |
-10.2% [95% CI: -23.4% - 3.8%], P=0.13 |
0.7% [95% CI: -8.3% - 9.8%], P=0.88 |
“IEO-TIRADS” vs. “EU-TIRADS” |
-16.5% [95% CI: -25.2% - -7.0%], P<0.001 |
26.5% [95% CI: 11.3% - 39.7%], P<0.001 |
-1.4% [95% CI: -9.7% - 6.9%], P=0.73 |
- In Discussion section, the author indicated the clinical performance of IEO-TIRADS was similar to other TIRADS system. It is recommended to discuss more advantages of IEO-TIRADS than EU-TIRADS and S-Detect, which strengthens the purpose of this study.
Thanks for the comment, we have modified the sentence “…We can conclude that “IEO-TIRADS” classification system for the evaluation of TN has a similar diagnostic performance to both “S-Detect” and “EU-TIRADS”. We believe that this new model makes the report easier to read and understand by surgeons and patients. The graphical representation maps of TN and cervical LN identified during the US examination facilitate the TN and LN location according to anatomical planes. This increases the diagnostic confidence in evaluating of the benign or malignant nature of a lesion in a rapid, easy and effective way…

Round 2
Reviewer 1 Report
Dear authors,
I only have minor comments.
Page 4, line90 : FNAB does not provide histological results. FNAB provides cytological results. Please correct this mistake and also check this issue in the whole manuscript.
Page 6, line 146 : Regarding, IEO-TIRADS system, we still not have the cut off used to classify the nodule as benign or malignant for the statistical analysis. What is the cut off used ? ≥4 (suspected ?), ≥5 (malignant) ?
It should be interesting to provide (in supplementary material for example) Se, Sp and DA according to different cut-off (with a ROC curve analysis).
I have no additional comments.
Author Response
Please see attachment
Reviewer 1
Review Report (Round 2)
Dear authors,
I only have minor comments.
Page 4, line90 : FNAB does not provide histological results. FNAB provides cytological results. Please correct this mistake and also check this issue in the whole manuscript.
Errors have been corrected.
Page 6, line 146 : Regarding, IEO-TIRADS system, we still not have the cut off used to classify the nodule as benign or malignant for the statistical analysis. What is the cut off used ? ≥4 (suspected ?), ≥5 (malignant)?
We have added a table with the cut off.
IEO-TIRADS classification |
IEO-TIRADS score |
Frequency |
Percent |
Benign |
1 (Very unlikely) |
9 |
6.43 |
Benign |
2 (Unlikely) |
44 |
31.43 |
Malignant |
3 (Doubt) |
23 |
16.43 |
Malignant |
4 (Suspected) |
26 |
18.57 |
Malignant |
5 (Malignant) |
38 |
27.14 |
It should be interesting to provide (in supplementary material for example) Se, Sp and DA according to different cut-off (with a ROC curve analysis).
We have added two table in supplementary material.
Table X. Sensitivity, specificity and diagnostic accuracy according to different cutoff of IEO-TIRADS score.
Cutoff |
Diagnostic performance |
||
Sensitivity [95% CI] |
Specificity [95% CI] |
Accuracy [95% CI] |
|
≥2 (Unlikely) |
98.9% [96.8% - 100%] |
16.3% [6.0% - 26.7%] |
70.0% [62.4% - 77.6%] |
≥3 (Doubt) |
73.6% [64.6% - 82.7%] |
59.2% [45.4% - 73.0%] |
68.6% [60.9% - 76.3%] |
≥4 (Suspected) |
59.3% [49.3% - 69.4%] |
79.6% [68.3% - 90.9%] |
66.4% [58.6% - 74.3%] |
≥5 (Malignant) |
38.5% [28.5% - 48.5%] |
93.9% [87.2% - 100%] |
57.9% [49.7% - 66.0%] |
Figure X. ROC curve of the model with IEO-TIRADS score as independent variable and histopathological result (Benign vs. Malignant) as dependent variable (Please see attachment)
I have no additional comments.

Reviewer 2 Report
All the issues were addressed.
Author Response
All the issues were addressed.
Thanks you.